# The Phenylpropanoid Gatekeeper CtPAL1 Coordinates ABA-Induced Flavonoid Biosynthesis and Oxidative Stress Tolerance in Safflower (*Carthamus tinctorius* L.)

**DOI:** 10.3390/plants14233606

**Published:** 2025-11-26

**Authors:** Xiaoyu Liu, Guanyao Zhang, Mingran Dai, Hong Zhao, Wei Ma, Yanli Hu, Na Yao, Jian Zhang, Naveed Ahmad, Xiuming Liu

**Affiliations:** 1College of Life Sciences, Engineering Research Center of the Chinese Ministry of Education for Bioreactor and Pharmaceutical Development, Jilin Agricultural University, Changchun 130118, China; l15141187176@163.com (X.L.);; 2Institute for Safflower Industry Research of Shihezi University/Pharmacy College of Shihezi University/Key Laboratory of Xinjiang Phytomedicine Resource and Utilization, Ministry of Education, Shihezi 832003, China; 3Agricultural-Forestry-Grassland and Ecological Environmental Protection Center of 161st Regiment, Tacheng 830091, China

**Keywords:** safflower, flavonoid synthesis, phenylalanine ammonia-lyase, abscisic acid

## Abstract

Phenylalanine ammonia-lyase (PAL) catalyzes the first committed step in the phenylpropanoid pathway that governs the entry of carbon flux into flavonoid biosynthesis and stress-responsive metabolism. However, how PAL explicitly mediates hormonal-induced flavonoid biosynthesis and promotes the antioxidant defense system in safflower (*Carthamus tinctorius* L.) remains largely unknown. Here, we functionally characterize CtPAL1 and demonstrated its regulatory role in abscisic acid (ABA)-induced flavonoid biosynthesis and antioxidant defense. Using phylogenetic and structural predictions, we found that CtPAL1 was placed within a conserved branch of Asteraceae PAL proteins. A promoter analysis indicated multiple hormone- and stress-responsive cis-elements, and exposure to abiotic and hormonal treatments elicited complex, stimulus-dependent dynamics of *CtPAL1* expression and flavonoid accumulation. Upon ABA treatment, the expression of *CtPAL1* is rapidly induced, triggering early flavonoid biosynthesis. Moreover, *CtPAL1*-overexpressing Arabidopsis lines exhibited enhanced tolerance to ABA-induced stress by lower lipid peroxidation and higher antioxidant enzyme activities, accompanied with increased flavonoid production. Importantly, the transgenic overexpression of *CtPAL1* in Arabidopsis led to the upregulation of the upstream flavonoid pathway genes (At4CL, AtCHI) and elevated total flavonoid levels (1.07–1.27-fold versus wild type), while silencing in safflower caused a reduced flavonoid content (0.52–0.77× controls) and the downregulation of pathway genes. A biochemical assay also confirms that recombinant CtPAL1 efficiently converts L-phenylalanine to trans-cinnamic acid, validating its catalytic function. Together, our results demonstrate that CtPAL1 functions as a highly conserved and functionally active PAL enzyme in safflower and acts as an ABA-responsive modulator of flavonoid biosynthesis and antioxidant defense.

## 1. Introduction

Safflower (*Carthamus tinctorius* L.) is a traditional medicinal and oilseed crop known for its rich repertoire of bioactive secondary metabolites, particularly flavonoids [1,2]. These compounds contribute not only to the plant defense, pigmentation, and signaling processes, but also to its recognized therapeutic potential in humans [3,4]. Despite extensive studies on the composition and pharmacological relevance of safflower flavonoids [5,6], the molecular mechanisms regulating their biosynthesis remain incompletely understood. Studies have attempted to functionally characterize the downstream flavonoid biosynthetic genes in safflower, for example—*CtFLS1* [7]; *CtCHI* [8], *CtCYP450s* [9,10,11], *CtHCT* [12,13], *CtC4H* [14], and *CtF3’5’H* [15]—and transcription factors such as *CtbHLH* [16], *CtbZIP* [17], *CtMYB* [18], and *CtDREB* [19]; however, the upstream entry-point enzyme phenylalanine ammonia-lyase (PAL), and, in particular, the functional roles of individual PAL isoforms in safflower remain unexplored. PAL is the rate-limiting gateway of the phenylpropanoid pathway and shows isoform-specific expression and regulatory behaviors in other species, implying that individual PAL homologs can differentially channel carbon flux toward flavonoids, lignin, and other phenolics [20]. Current developments in the genome-wide and association analysis revealed that PAL gene families possess significant evolutionary diversification, suggesting the need for isoform-level functional studies in order to bridge sequence-level divergence, transcriptional activities, and metabolic turnover [21].

Phenylpropanoid metabolism in plants is initiated by the action of the PAL enzyme, which catalyze phenylalanine into trans-cinnamic acid [22,23]. Further modifications are carried out by cinnamate-4-hydroxylase (C4H) and 4-coumarate:CoA ligase (4CL) enzymes, directing diverse phenylpropanoid branches to produce distinct classes of flavonoids, lignins, and stilbenes [24,25,26]. As a pivotal metabolic enzyme, PAL modulates pathways relevant to plant stress responses and holds promise for therapeutic applications in phenylketonuria and oncology [27,28]. Since its first report in 1961 [29], PAL has shown adaptive responses to both biotic and abiotic stress along with exogenous hormones [30,31]. Comparative studies in pear [32], rice [33,34], tobacco [35], and raspberry [36] have shown the functional diversity of PAL enzymes. In safflower, a total of six PAL genes have been identified [36], with CtPAL1–CtPAL5 retaining conserved catalytic sites and domains commonly associated with the PAL family. Nevertheless, the functional roles of CtPALs in regulating safflower flavonoid biosynthesis and stress responses remains unresolved.

Global crop losses from abiotic stresses such as drought, salinity, and heat or cold extremes emphasize the need to uncover the molecular basis of plant resilience [37]. To withstand these adverse environmental stresses, plants mainly rely on physiological, hormonal, and biochemical responses [38,39]. Of the hormonal network, abscisic acid (ABA) serves as a pivotal mediator in modulating the growth, development, and abiotic-stress signaling pathways [40]. For instance, ABA-induced transcriptional and physiological changes, such as stomatal closure, support water-saving strategies and the homeostatic balance in stressed plants [41,42]. Apart from its primary actions, ABA also coordinates with secondary metabolic switches, including the phenylpropanoid pathway, to improve plant tolerance to stress [31,43,44,45]. Given the pivotal role of the PAL enzyme as the entry point to the flavonoid pathway, it is pragmatic to investigate the PAL response to ABA and pave the way to understanding the underlying molecular mechanisms of ABA-mediated flavonoid synthesis and the antioxidant defense system in plants.

In this study, we identify and functionally characterize CtPAL1, an ABA-responsive PAL isoform from safflower that regulates flavonoid biosynthesis and antioxidant defense system. To elucidate its function, we initially carried out structural predictions, and phylogenetic analyses, and then use the overexpression and virus-induced gene silencing (VIGS) approaches. Our findings from a comparative biochemical, physiological, and transcriptional and enzymatic analysis collectively suggested CtPAL1 acts as a conserved, functionally active PAL enzyme that mediates the ABA-responsive regulation of flavonoid production and antioxidant activity in safflower.

## 2. Results

### 2.1. Isolation, Evolutionary Insights, and Structural Prediction of CtPAL1

We begin with the cloning and isolation of the full-length *CtPAL1* gene from the safflower genome by employing PCR amplification. After confirmation with Sanger sequencing, we constructed the plant overexpression vector pCAMBIA3301 for generating transgenic plants and the subsequent functional analysis. Prior to isolation, we investigated the physicochemical properties of CtPAL1 and it showed that the 2124 bp open reading frame of CtPAL1 encodes a 76.8 kDa protein. The isoelectric point was 5.84 with an instability index of 34.99, suggesting that the predicted protein is stable. Similarly, the high aliphatic index (91.77) along with the negative GRAVY value (–0.153) shows that it is a hydrophilic protein, compatible with cytosolic solubility (Appendix A).

We then carried out phylogenetic analysis to examine the evolutionary relationship of CtPAL1. The results suggested that CtPAL1 was clustered together in a distinct branch of PAL proteins within the phylogenetic tree generated from different Asteraceae species, including artichoke, gerbera, snow lotus, and Cirsium (Figure 1A). This indicated that *CtPAL1* contained significant conservation and functional characteristics in line with those other PAL proteins from the Asteraceae species. Consistent with this, multiple sequence alignment showed >80% amino acid identity between CtPAL1 (accession MF421795.1) and PAL proteins from *Salvia miltiorrhiza* (ShPAL), *Lactuca sativa* (LsPAL), *Populus tomentosa* (CoPAL), *Cirsium japonicum* (CjPAL), *Saussurea involucrata* (SiPAL), *Andrographis paniculata* (ApPAL), *artichoke* (CcPAL), and *Eupatorium adenophorum* (AaPAL) (Figure 1B). Moreover, the structural features of CtPAL1 showed that it consists of 705 residues, rich in leucine (10.9%), alanine (9.2%), and glycine (8.3%), with minor residues from tryptophan (0.6%) and cysteine (1.1%) (Appendix A). This distribution of amino acid pattern revealed moderate nonpolar enrichment, suggesting a stable, compact tertiary structure that exhibits canonical PAL folds (Figure 2A–D). Together, these results establish CtPAL1 as a highly conserved PAL protein in safflower, likely maintaining canonical functions in phenylpropanoid and flavonoid biosynthesis.

### 2.2. Basal Expression Patterns of CtPAL1 Reveal Tissue-Specific but Nonlinear Correlation with Flavonoid Accumulation Under Normal Condition

The basal expression pattern of *CtPAL1* was investigated in different tissues of safflower, including the roots, stems, and leaves under normal growth conditions. Using the root tissues as the basal reference (1.0), the *CtPAL1* transcripts were most abundant in the stem tissues, whereas the expression was suppressed in the leaves (Figure 3). Under simultaneous growth conditions, the total flavonoid content across the selected safflower tissues indicated diverse accumulation, with leaves exhibiting the highest accumulation, followed by stems and roots (Figure 3). Meanwhile, *CtPAL1* demonstrated a higher expression in the stem; however, flavonoid accumulation was evidently greater in the leaves, suggesting a complex and nonlinear correlation. This is because of the multifaceted regulation of the phenylpropanoid pathway, with *CtPAL1* governing the systematic phenylalanine flux into distinct downstream branches. The distribution of this relative metabolic flux through different branches of the phenylpropanoid pathway is likely regulated in a tissue-dependent manner and may correspond to specific physiological factors beyond *CtPAL1* expression alone. Moreover, external environmental cues such as phytohormones and abiotic stresses can further modulate this regulatory coordination. Such incidents were also found in earlier studies. For instance, the garlic roots showed a higher *PAL* expression, whereas an increased flavonoid content was found in the bulbs [46]. From these findings, we inferred that *CtPAL1* exhibits tissue-specific expression under basal conditions, suggesting a non-linear correlation with flavonoid synthesis and that it may act as a conditional rather than constitutive regulator.

### 2.3. Abiotic Stress and Hormonal-Induced CtPAL1 Expression Suggested Modulation in Flavonoid Accumulation

In order to investigate the possible link between *CtPAL1*-induced expression and flavonoid accumulation in safflower, we first carried out a promoter analysis and identified an array of cis-elements responsive to hormones and abiotic stress. Noticeably, cis-elements such as ABA, methyl jasmonate (MeJA), salicylic acid (SA), gibberellin (GA), light, and defense-related signals (Figure 4) were found in the *CtPAL1* promoter, indicating that *CtPAL1* expression can be finely tuned by a variety of environmental cues. Next, an expression analysis of *CtPAL1* and quantification of the flavonoid content was carried out under abiotic stresses (drought, salt, and UV-B) and hormonal treatments (ABA, GA, SA, and MeJA) over different time points (Figure 5).

Simulated drought stress (DS-1) led to a steady increase in total flavonoid levels, with a maximum spike (~15 mg/g FW) at day 14. At the same time, *CtPAL1* expression was also upregulated, respectively, except at day 7. On the other hand, natural drought stress (DS-2) resulted in a patronized increased (~10 mg/g FW) in flavonoid accumulation at 24 h and 14 d, with *CtPAL1* transcript levels maximized at day 3. Likewise, salt stress induced the accumulation of flavonoids up to ~11 mg/g FW at day 7, with the highest (~6-fold) *CtPAL1* expression at the 12 h time point. Furthermore, exposure to ABA induced a gradual spike in flavonoid levels, with the highest levels (~15 mg/g FW) observed at the 24 h time point, and *CtPAL1* transcript abundance at the 3 h and 12 h time points, respectively. In contrast, GA application induced a continuous increase in flavonoid content (~22 mg/g FW) over 48 h, whereas the expression level of *CtPAL1* showed moderate changes (~1.0–2.0-fold) over different time points. Another time-course analysis likewise showed a continuous flavonoid accumulation pattern alongside a transient, early peak in *CtPAL1* expression (Figure 5). Under SA, MeJA, and UV-B conditions, the accumulation of flavonoids followed a diverse temporal pattern, with *CtPAL1* transcript abundance frequently preceding or only partially reflecting flavonoid accumulation. For instance, *CtPAL1* demonstrated a transient induction under SA and MeJA, while UV-B induced a constant flavonoid accumulation with slight changes in CtPAL1 expression. From these findings, we implied that ABA shows the most significant and coordinated hormonal signal in establishing a synchronized pattern between *CtPAL1* expression and flavonoid accumulation compared to the control. Thus, we chose ABA for further comparative investigations to explicate the molecular link between ABA-triggered *CtPAL1* expression and flavonoid accumulation.

### 2.4. Analysis of CtPAL1 Expression and Its Impact on Flavonoid Pathway-Related Genes in Transgenic Arabidopsis

Genomic DNA was extracted from Arabidopsis leaves, and a total of 21 T_3_ transgenic lines harboring *CtPAL1* were successfully identified (Figure 6A). The results of the *CtPAL1* expression level investigated through a qRT-PCR analysis confirmed that four transgenic lines, namely, OE10, OE12, OE16, and OE18, demonstrated a significantly higher *CtPAL1* expression than the wild type (WT). Noticeably, the two lines named OE10 and OE16 stand out as the high-expression lines and, therefore, were selected for subsequent investigation (Figure 6A–B). We further investigated the expression level of key flavonoid biosynthetic genes (*At4CL AtCHI, AtC4H,* and *AtANS*) in the two OE10 and OE16 high-expression lines to confirm the effect of *CtPAL1* on their transcript abundance. The qRT-PCR results suggested that the two downstream pathway genes *At4CL* and *AtCHI* showed a significant upregulation in transgenic lines when compared to WT (Figure 6). In contrast, the other two genes *AtC4H* and *AtANS* demonstrated a reduced expression than the WT (Figure 6). This reduced expression of *AtC4H* might arise from the feedback inhibition in the flavonoid pathway or the absence of sufficient transcriptional regulation. It is also plausible that transcripts for the intermediate steps are less and/or not stable, which led to a suppressed expression compared to both the upstream (*AtPAL*, *At4CL*) and downstream (*AtCHI*, *AtANS*) genes. Taken together, these results imply that *CtPAL1* overexpression appears to regulate the flavonoid biosynthetic pathway at the initial branch point, while potentially imposing certain downstream steps.

### 2.5. CtPAL1 Overexpression Significantly Modulates Flavonoid Accumulation in Transgenic Arabidopsis

To further delineate the regulatory effect of *CtPAL1* in directing the metabolic flux of the flavonoid pathway, we investigated the accumulation pattern of the total flavonoid content in the two high-expression lines of *CtPAL1*-overexpressing transgenic lines for flavonoid quantification. The quantification analysis showed a consistent increased flavonoid level in transgenic lines relative to WT (Figure 7). Approximately, a ~1.07–1.27-fold increase was observed in the accumulation of flavonoids in both the OE-10 and OE-16 transgenic lines (Figure 7). This rise in the flavonoid content in the transgenic lines implies that *CtPAL1* overexpression may cause a significant induction in the biosynthetic machinery of the flavonoid pathway, pinpointing the role of CtPAL1 as a positive regulator. It is possible that CtPAL1 likely increases the availability of cinnamic acid and of the upstream precursors, which further facilitate the flux toward flavonoid end products. On the other hand, despite the enhanced accumulation of flavonoids, no major phenotypic differences were observed between the transgenic overexpression lines and WT plants, suggesting that *CtPAL1* overexpression specifically influences metabolic regulation without perturbing the overall growth or developmental patterns. The elevated flavonoid levels may also contribute to improved physiological resilience, as flavonoids play vital roles in ROS scavenging, and defense responses.

### 2.6. CtPAL1 Silencing Leads to Impaired Flavonoid Accumulation and Altered Expression of Flavonoid-Pathway-Related Genes in Transgenic Safflower

To provide further genetic evidence on the role of the *CtPAL1* gene as a positive regulator of safflower flavonoid biosynthesis, we employed a TRV (tobacco rattle virus)-mediated gene-silencing method known as VIGS to transiently suppress *CtPAL1* activity in planta. In total, four VIGS-mediated *CtPAL1*-suppressed safflower lines (henceforth referred to as mutant) were obtained using the TRV 1 + TRV 2 connected vector as the control, and TRV1 + pTRV2-*CtPAL1* as the mutant group (Figure 8A). Compared to the control (harboring a TRV 1 + TRV 2 empty vector), the four *CtPAL1* mutant lines indicated significant reductions (~0.52–0.77 times) in total flavonoid content compared to that of the control group (Figure 8A). Importantly, the phenotype of the *CtPAL1* mutant lines also showed slightly inhibited growth and/or delayed development after two weeks of injection when compared to the control plants. An additional layer of confirmation was carried out by analyzing the expression level of *CtPAL1* and four selected key flavonoid biosynthetic genes (*Ct4CL CtCHI, CtC4H,* and *CtANS*) in the mutant and control lines of safflower using qRT-PCR analysis. The expression results clearly shown a significant downregulation of *CtPAL1* in the mutant line compared to the control line, suggesting a partial loss of function (Figure 8B). Furthermore, the expression levels of the four key flavonoid biosynthetic genes also demonstrated varying degrees of downregulation in the mutant line compared to the control line. Together, these findings affirm that the loss of *CtPAL1* function in the mutant lines led to an reduced expression of other key genes and flavonoid content, thereby supporting CtPAL1 as a key candidate in regulating flavonoid biosynthesis.

### 2.7. CtPAL1 Overexpression Enhances ABA-Induced Antioxidant Defense by Modulating Flavonoid Biosynthesis in Arabidopsis

Given its central role upstream in the phenylpropanoid and flavonoid biosynthetic pathways, *CtPAL1* was selected for functional characterization under ABA application. The overexpression of the Arabidopsis line *CtPAL1*-OE16 was used to evaluate ABA-induced metabolic regulation and the antioxidant defense system. For this purpose, the total flavonoid content in *CtPAL1*-OE16 and WT plants under ABA application over different time points was analyzed. The results showed that, in *CtPAL1*-OE plants, the total flavonoid content was initially decreased and then increased at the 3–12 h time points, respectively, when compared with the WT plants (Figure 9A,B). The temporal expression level of *CtPAL1* and endogenous *AtPAL1* was compared under ABA treatments over different time points. It was observed that the expression pattern of *CtPAL1* followed a differential pattern, demonstrating an initial suppression and then an increase to its maximum at 24 h after ABA application (Figure 9C), consistent with the changes observed in flavonoid content. On the other hand, *AtPAL1* showed an opposite expression trend than *CtPAL1*, demonstrating downregulation at the 24 h and 36 h time points. Moreover, significantly lower levels of MDA were also observed in the *CtPAL1*-OE transgenic plants across a 0–48 h ABA treatment period compared to WT plants (Figure 9D). This indicates the activation of an effective ROS detoxification system in the transgenic plants, leading to an enhanced antioxidant defense system when encountering ABA elicitation. Further investigation was carried out on the expression changes of flavonoid pathway genes including *AtC4H*, *At4CL*, *AtCHI*, and *AtANS* using the *CtPAL1*-overexpression lines and WT plants under ABA application at different time points. The expression data by qRT-PCR suggested that both *AtC4H* and *At4CL* showed a significant downregulation at 24 h and 12 h, respectively, in transgenic plants when compared with WT. However, the transcript abundance of *AtCHI* and *AtANS* showed relatively no significant changes or only minor changes (Figure 10). From these findings, we implied that *CtPAL1* overexpression significantly promotes ABA-induced flavonoid accumulation, thereby modulating the enhanced antioxidant defense system.

### 2.8. CtPAL1-Overexpressed Transgenic Plants Under ABA Showed Enhanced Antioxidant Defense by Reducing Oxidative Damage

As described previously that *CtPAL1*-overexpressing transgenic plants exhibit enhanced flavonoid biosynthesis when exposed to ABA and the fact that ABA is a positive inducer of ROS accumulation, we speculated that *CtPAL1* may be involved in enhancing the antioxidant defense system in response to ABA exposure. To investigate this, we compared the key antioxidant enzymes such as superoxide dismutase (SOD), catalase (CAT), and peroxidase (POD) activities in *CtPAL1*-overexpressing transgenic and WT plants under ABA treatment. We first investigated the SOD enzymatic activities and the results suggested that CtPAL1-overexpressing (OE-PAL) transgenic lines showed significantly higher SOD activities than those of the WT plant after ABA treatments. As shown in Figure 11A, SOD enzymatic activity was continuously elevated in *CtPAL1*-overexpressing transgenic plants at the 0–48 h time points, reaching to its maximum at 12 h, in comparison to WT plants. This significant increase in SOD activity indicated a more robust dismutation of superoxide radicals (O_2_^−^) into hydrogen peroxide (H_2_O_2_), which is considered the first line of defense during ROS detoxification. Similarly, ABA treatment significantly induced CAT enzymatic activities in *CtPAL1*-overexpressing transgenic plants but with a variable pattern compared to those of the WT plants. It was found that CAT activity was the relatively highest at 6 and 36 h in transgenic plants when compared with the WT plants (Figure 11B). The significant increase in POD enzymatic activities may further accelerate the decomposition of H_2_O_2_ into water and oxygen. Last but not the least, POD enzymatic activities were also significantly induced by ABA in *CtPAL1*-overexpressing transgenic plants with a continuous pattern over the 3–48 h time points, peaking at the 3 h and 12 h time points, respectively, when compared to WT plants (Figure 11C). These increased POD activities may facilitate an additional layer of H_2_O_2_ reduction through the oxidation of phenolic substrates. Together, these improved ROS-scavenging enzyme activities is in line with our previous findings of the reduced MDA content. and the detectable increase in flavonoid content (Figure 9) implied that *CtPAL1* is likely regulating both the enzymatic- and non-enzymatic-based antioxidant defense under ABA treatment.

### 2.9. Protein Expression and In Vitro Enzymatic Activity of CtPAL1

The recombinant CtPAL1 protein was heterologously expressed in *Escherichia coli* cells using the in vitro enzymatic assays in order to validate its catalytic activity. At first, the recombinant CtPAL1 protein was efficiently expressed in competent *E. coli* cells and was detected on SDS-PAGE (Figure 12A). The expressed CtPAL1 protein was purified, yielding more than 90% of the total protein fraction, providing a reliable basis for the subsequent in vitro enzymatic assay. The subsequent enzymatic activity assays showed that the CtPAL1 protein efficiently catalyzed the deamination of L-phenylalanine (L-Phe) to produce trans-cinnamic acid (Figure 12B). The presence of trans-cinnamic acid fractions within the reaction mixture after the CtPAL1 addition strongly suggested that the CtPAL1-encoding enzyme possesses typical PAL activity, since the conversion of L-phenylalanine (L-Phe) to produce trans-cinnamic acid represents the first committed step in the phenylpropanoid pathway, involving the conversion of primary amino acid metabolism into the biosynthesis of a variety of secondary metabolites, such as flavonoids, lignins, and other phenolic compounds. From these preliminary findings, we inferred that CtPAL1 is a functionally active enzyme exhibiting strong catalytic activity, and thereby participating in a central role in mediating phenylpropanoid biosynthesis in safflower.

## 3. Discussion

Flavonoids and other polyphenols are known to act as natural antioxidants providing both enzymatic and non-enzymatic channels for ROS detoxification [47,48]. These naturally occurring compounds can directly neutralize ROS, such as superoxide anions, hydrogen peroxide, and hydroxyl radicals, by using enzymatic degradation, which help in protecting cells from oxidative stress, and regulate redox homeostasis under fluctuating conditions [47,49,50,51]. The efficiency of these compounds as antioxidants is attributed to their structural configuration, enabling them to donate electrons or protons to inhibit ROS production [52,53]. On the other hand, phenolic compounds can also offer enzymatic channels acting as electron donors for guaiacol-type peroxidases (for instance, horseradish peroxidase), aiding in the reduction of hydrogen peroxide into water molecules [48,54,55]. This phenomenon is parallel to the ascorbate peroxidase (APX) mechanism, in which ascorbate acts as an electron donor. During stress conditions, the ascorbate levels are significantly reduced—this is when phenolics play its role as alternative substrates for the peroxidase-mediated detoxification of H_2_O_2_ [48]. Flavonoid-induced oxidation driven by peroxidases produces phenoxyl radicals, which are recycled by systems such as monodehydroascorbate reductase, contributing to an integrated and resilient antioxidant network [54]. The present study was designed to investigate the functional role of CtPAL1, a phenylalanine ammonia-lyase from safflower, and elucidate its role in ABA-mediated antioxidant defense via flavonoid biosynthesis.

### 3.1. CtPAL1 Functions as Catalytic Gatekeeper in the Flavonoid Pathway

Using conservation and evolutionary analysis, the candidate CtPAL1 showed high structural conservation and a greater than 80% amino acid identity among Asteraceae PAL enzymes, from *S. miltiorrhiza* and *L. sativa* (Figure 1). The observed phylogenetic conservation and identity with other PAL members suggests that CtPAL1 retains the core features of PAL enzymes and that it may participate in an essential role in the flavonoid biosynthesis across plant lineages [56,57,58]. Unlike other safflower PAL isoforms, *CtPAL1* demonstrated the tissue-specific and non-linear regulation of flavonoid content under basal conditions; however, it shows an ABA-induced enhanced expression, leading to improved flavonoid accumulation and antioxidant defense (Figure 9, Figure 10 and Figure 11). A similar pattern of similar findings was reported in garlic roots, which showed a higher PAL expression in the roots, whereas the bulb tissues contain an increased flavonoid content [48]. Similarly, the expression pattern of four *PAL* genes along with their enzymatic activities demonstrated a significant increase in the flavonoid content when the lettuce stems were challenged with external signals such as wounding [59]. These findings led us to the conclusion that CtPAL1 acts as a gatekeeper that directed various downstream phenylalanine branches when induced with ABA treatment. Further evidence was presented with *CtPAL1* silencing effects, which led to the noticeable suppressed accumulation of flavonoid content and the expression of key flavonoid biosynthetic genes, whereas its overexpression showed an opposite trend, resulting in elevated flavonoid biosynthesis (Figure 8, Figure 9 and Figure 10).

### 3.2. ABA as Key External Signal for Modulation of CtPAL1-Mediated Flavonoid Biosynthesis and Antioxidant Defense

Of all the tested hormonal and abiotic treatments, we found that ABA treatment provides considerable evidence and a regulatory link between CtPAL1 induction and the enhanced flavonoid biosynthesis and antioxidant system. This led us to the conclusion that CtPAL1 is likely an ABA-responsive regulator (Figure 5). The role of ABA in inducing the expression of genes that are actively involved in secondary metabolite biosynthesis and antioxidant production is well-documented as well [60]. Noticeably, studies also suggested that ABA signals can effectively enhance PAL activity, which ultimately resulted in improved flavonoid accumulation [61,62]. From these findings, it is well-studied that ABA coordinates with flavonoid biosynthesis, which further activates the non-enzymatic channels of the antioxidant defense system in plants to cope with adverse environmental constrains [58,63]. During our investigation, we found that *CtPAL1*-overexpressing transgenic Arabidopsis when exposed to ABA treatment resulted in increased flavonoid accumulation levels compared to the WT plants. Meanwhile, the MDA content also showed a significant reduction whereas the activities of SOD, CAT, and POD, key antioxidant enzymes, showed an improved pattern in transgenic plants compared to WT (Figure 9, Figure 10 and Figure 11). These findings confirmed that ABA-induced CtPAL1 is actively involved in enzymatic as well as non-enzymatic antioxidant defenses in safflower. Previous studies showed similar results where such enhanced antioxidant enzyme activities are being documented to show reduced oxidative stress in other plant species [64,65]. Other works of research have argued that the transient increase in flavonoid levels might have occurred due to the increased ROS levels when plants encounter stress-induced oxidative damage [66,67]. Another possibility is the synergy between the enzymatic and non-enzymatic channels of the antioxidant defense system, which finetune a more robust defense response when encountering such oxidative damage. Consistent with this, several studies showed that the activation of robust antioxidant channels via both enzymatic and non-enzymatic activities is crucial for maintaining plant growth and survival under stress conditions [65,68]. Hence, the identification of CtPAL1 as an important regulator within this framework not only highlights its functional capacity but also suggests future strategies for enhancing stress resistance in crops through targeted breeding or genetic engineering strategies.

### 3.3. Crosstalk Between CtPAL1 and Downstream Flavonoid Pathway Genes

In our investigation, the overexpression of *CtPAL1* showed a significant alteration in the expression level of *C4H*, *4CL*, *CHI*, and *ANS*, which are the key flavonoid biosynthetic genes within the phenylpropanoid–flavonoid network. The pattern we observed—the upregulation of 4CL and CHI with the concurrent downregulation of C4H and ANS—is best interpreted as a reconfiguration of metabolic flux rather than as a simple, linear up- or downregulation of the whole pathway (Figure 8 and Figure 10). In other words, CtPAL1 activity appears to change the relative allocation of phenylalanine-derived carbon into competing branches of the pathway, favoring certain flavonoid branches (early-branch enzymes such as 4CL/CHI) while reducing flux into others (C4H/ANS). This kind of re-routing is concordant with two related mechanistic frameworks: (i) transcriptional network regulation, where master transcription factors (e.g., R2R3-MYBs, and bHLHs, sometimes acting in MBW complexes) selectively activate distinct sets of pathway genes to produce specific flavonoid subclasses [69,70]; and (ii) metabolic channeling/metabolon formation, where physical enzyme–enzyme interactions (for example, a PAL–C4H couple) and local substrate channeling determine the efficiency and destination of intermediate metabolites [71,72,73].

Two non-exclusive mechanistic explanations likely account for our expression signatures. First, we have the feedback/feed-forward regulation via metabolite pools: changes in the pool sizes of cinnamate, p-coumarate, or other intermediates caused by altered PAL activity can feed back onto transcriptional regulators or lead to substrate competition that reduces the apparent activity (or transcriptional stability) of certain branches. This hypothesis is consistent with reports showing that perturbations in the early steps of the pathway change the downstream gene expression and product profiles in a branch-specific manner [74,75,76]. Second, coordinated transcriptional control such as ABA or stress-activated TFs (e.g., AREB/ABF family bZIPs) can directly regulate both CtPAL1 and downstream biosynthetic genes [77,78], while MBW complexes (MYB-bHLH-WD40) tune the branch specificity (anthocyanins, PAs, and flavonols) [79]. Under this model, CtPAL1 acts as an ABA-sensitive node whose induction is integrated with TF programs that differentially modulate C4H, 4CL, CHI, and ANS, producing the observed asymmetric expression changes. Evidence that ABA signaling reprograms flavonoid gene networks under stress supports this scenario [74,80,81]. Furthermore, the physical organization of enzymes into transient metabolons can impose directionality on flux independent of the bulk transcript levels: PAL–C4H interactions have been shown to favor channeling into certain branches, and differences among PAL paralogs in their ability to enter such complexes can, therefore, alter downstream outcomes. If CtPAL1 preferentially participates in metabolons that either exclude or deprioritize ANS-related branches, this could help explain why ANS is downregulated despite the increased upstream PAL activity.

In conclusion, our results identified CtPAL1 as a potential ABA-responsive candidate enzyme that bridged flavonoid biosynthesis and improved antioxidant defense in safflower. The ability of CtPAL1 to induce flavonoid accumulation, its crosstalk with key downstream flavonoid pathway genes, and the improved ROS-scavenging activities both at the enzymatic and non-enzymatic levels under ABA conditions underscore its essential role in maintaining plant growth and survival. Future studies involving transcription-factor-binding assays, metabolic flux analysis, and protein–protein interaction studies will be contributory in uncovering the precise regulatory mechanisms by which CtPAL1 orchestrates phenylpropanoid pathway responses to ABA and environmental stimuli.

## 4. Materials and Methods

### 4.1. Plant Materials, Treatments, and Growing Conditions

Seeds of safflower (*Carthamus tinctorius* L., Jihong No. 1) were soaked for 3–6 h to remove floating seeds, and then evenly sown in plastic pots (20 cm × 25 cm) containing a black soil:vermiculite mixture (7:3, *v*/*v*). Seedlings were grown in an artificial climate chamber at Jilin Agricultural University. Three stress treatments were applied to three-week-old uniform seedlings: ABA (100 µM, Solarbio, Beijing, China), methyl jasmonate (MeJA, 100 µM, Solarbio, Beijing, China), and cold stress (4 °C). For virus-induced gene silencing (VIGS) of CtPAL1, pTRV2-CtPAL1 suspension was injected into the abaxial side of leaves using disposable medical injectors. After injection, plants were incubated in darkness for 24 h and then under light for one week.

*Arabidopsis thaliana* seeds were germinated on MS medium (Solarbio, Beijing, China), and, later, transplanted into soil mixture (vermiculite:good soil = 7:3, *v*/*v*). Seedlings of WT, CtPAL1-overexpressing lines, and CtC4H1-overexpressing lines were treated with ABA stress once plants reached four weeks of age. Leaf samples were collected at early-stage intervals every 3 h and, later, every 12 h for up to 48 h. All samples were immediately frozen in liquid nitrogen and stored at –80 °C until further use.

### 4.2. Protein Structure Analysis

ExPASy ProtParam tool Version 3 (http://web.expasy.org/protparam (accessed on 10 September 2024)) was used to analyze the physicochemical properties of the protein encoded by *Carthamus tinctorius* CtPAL1 gene. The protein-conserved domain was analyzed by NCBI database CD-Search. Prot Scale and TMHMM Server 2.0 were used to analyze the hydrophilic and hydrophobic domains of the protein. SWISS-MODEL (https://swissmodel.expasy.org/) was used to predict the tertiary structure of CtPAL1 protein and carry out homology modeling.

### 4.3. Multiple Sequence Alignment and Phylogenetic Analysis

For multiple sequence alignments, NCBI blast search (https://BLAST.ncbi.nlm.nih.gov (accessed on 1 January 2024)) and TAIR website (http://www.arabidopsis.org/ (accessed on 1 January 2024)) were used. Using MEGA 11.0 software, plant species selected for PAL amino acid sequence alignment included sage, lettuce, *Populus tomentosa*, *Thistle*, *Snow Lotus*, *Andrographis paniculata*, *Artichoke*, and *Eupatorium adenophorum*. Then, a phylogenetic tree method was constructed based on adjacent linkage relationships and bootstrap was set to test 1000 replicates.

### 4.4. Construction of Prokaryotic Expression Vector of CtPAL1 Gene and Determination of Enzymatic Activity

The cloned fragment digested by restriction enzyme was ligated with linearized pGEX-KG vector and transformed into *E. coli*-competent state. After Amp+ resistance screening, PCR amplification and double enzyme digestion verification, positive recombinant colonies were determined. The positive recombinant plasmid was transformed into *E. coli* BL-21, inoculated into 2 mL LB medium containing Amp+ antibiotics, and cultured overnight at 37 °C. When OD600 value was about 0.4–0.5, IPTG of final concentration 0.6 mmol·L-1 was added, temperature was set at 16 °C, culture time was set at 40 h, and protein expression was induced. Take 50 mL of bacterial solution, centrifuge at 9500 r/min for 10 min, discard supernatant, resuspend bacteria in 5 mL 1 × PBS (pH 7.4) buffer solution, repeatedly freeze and thaw, sonicate, centrifuge at 12,000 r/min for 15 min, and take all of the supernatant. Afterwards, 2 mL of bacterial solution was added with 2.5 mL of 50 mmol/l phenylalanine solution and 0.5 mL of 0.05% CTAB solution. The reaction solution was incubated at 30 °C for 20 min and terminated by adding 0.3 mL of 6 mol/L HCI solution. Supernatant was detected by UV spectrophotometer at 278 nm. Enzyme activity (U) Definition 9: The amount of enzyme required to convert 1 μmol phenylalanine per minute to cinnamic acid.

### 4.5. Generation of T3 Transgenic Arabidopsis and Screening of High-Expression Lines

T2 generation transgenic Arabidopsis thaliana lines obtained previously were screened. Plants were treated with 0.1% glufosinate for selection, and genomic DNA was extracted from leaves using a plant genome extraction kit. Transgene integration was confirmed by PCR using CtPAL1 and BAR primers. Total RNA was then extracted from T3 generation plants, and qRT-PCR analysis was performed to identify lines with high expression of CtPAL1. All subsequent stress treatments were conducted using the selected T3 high-expression transgenic lines.

### 4.6. RNA Isolation and Real-Time Fluorescence Quantitative PCR Analysis

Samples frozen at –80 °C were quickly ground in liquid nitrogen and total RNA extracted using RNAiso Plus reagent (TIANGEN Biotech, Beijing, China). The concentration and quality of extracted RNA were determined using NanoDrop 2000 UV spectrophotometer (TaKaRa, Beijing, China) and gel electrophoresis. Reverse-transcription of cDNA templates was prepared using PrimeScript RT kit (Stratagene, San Diego, CA, USA). Then, qRT-PCR assay was performed using SYBR Premix Ex TaqII (TaKaRa, Kusatsu, Japan). Expression levels were normalized using the expression level of the 18s ribosomal RNA gene as an internal reference gene. The relative expression level was calculated using statistical formula of 2^−∆∆Ct^ method.

### 4.7. Determination of Total Flavonoid Content and Physiological and Biochemical Indices

The quantification of total flavonoid content in Arabidopsis and safflower was determined as follows: 0.1 g of plant tissue was ground in liquid nitrogen to a fine powder and transferred to a 10 mL centrifuge tube. Methanol (0.5 mL) was added, followed by 0.15 mL of 5% sodium nitrite solution and incubation for 6 min. Next, 2 mL of 4% sodium hydroxide solution and 0.15 mL of 10% aluminum nitrate solution were added sequentially, with a 6 min incubation period after each addition. The addition of distilled water makes up the final volume to 5 mL, and then the resultant mixture was carefully shaken and allowed to stand for 3 min at the least. Then, the samples were subject to centrifugation at 12,000 rpm for a period of 10 min. Finally, an equal volume of 0.3 mL of the supernatant was measured at 508 nm using a spectrophotometer. Flavonoid content was calculated from a rutin standard curve. Finally, the SOD, CAT, POD, activities, and quantification of MDA content were analyzed using Solarbio assay kits following the manufacturer’s protocol.

### 4.8. Statistical Analysis

The statistical calculation and analyzed data were shown as mean ± standard deviation using values obtained from three replicates. Most of the analyses were carried out using GraphPad prism v.9.1.3. The normality and variance tests were applied to each data set, and then the data were analyzed by analysis of variance (ANOVA) and other parametric tests. In case of ANOVA analysis, if the test statistic was significant, Tukey’s multiple post-comparison test was performed.

## 5. Conclusions

This study provides direct evidence that ABA-induced CtPAL1 plays a central role in flavonoid biosynthesis and antioxidant defense regulation in safflower. Through overexpression in Arabidopsis and VIGS in safflower, we demonstrated that CtPAL1 modulates the expression of key flavonoid biosynthetic genes and promotes the accumulation of flavonoid metabolites under ABA treatment. The improved flavonoid accumulation and enhanced antioxidant defense in transgenic lines highlights CtPAL1 as a pivotal upstream regulator in the flavonoid pathway when exposed to ABA. These findings not only deepen our understanding of the regulatory network of PALs but also provide potential strategies for metabolic engineering and stress resilience in plants.

## Figures and Tables

**Figure 1 plants-14-03606-f001:**
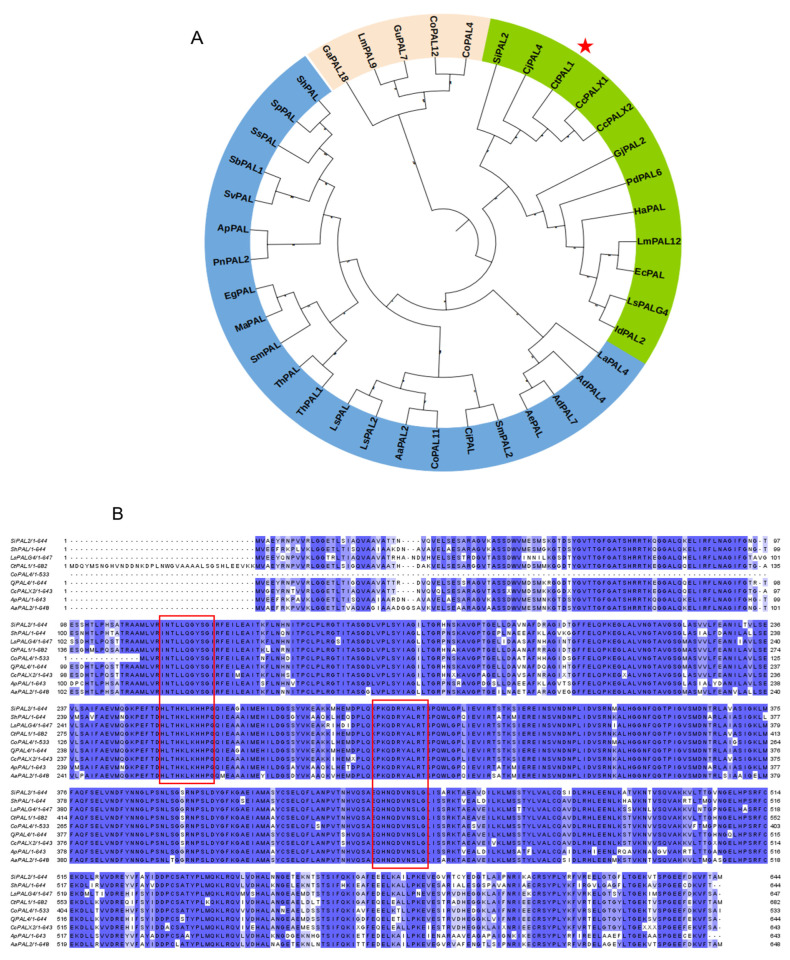
Phylogenetic analysis and multiple sequence alignment of PALs from safflower and other related plant species. (**A**) Phylogenetic tree of PAL members from safflower and artichoke, gerbera, snow lotus, and Cirsium. The tree was constructed with MEGA 11.0 using the Neighbor-Joining (NJ) method and 1000 bootstrap replications. (**B**) Multiple sequence alignment of CtPAL1 and PAL from other plant species. Conserved domains are highlighted in red boxes.

**Figure 2 plants-14-03606-f002:**
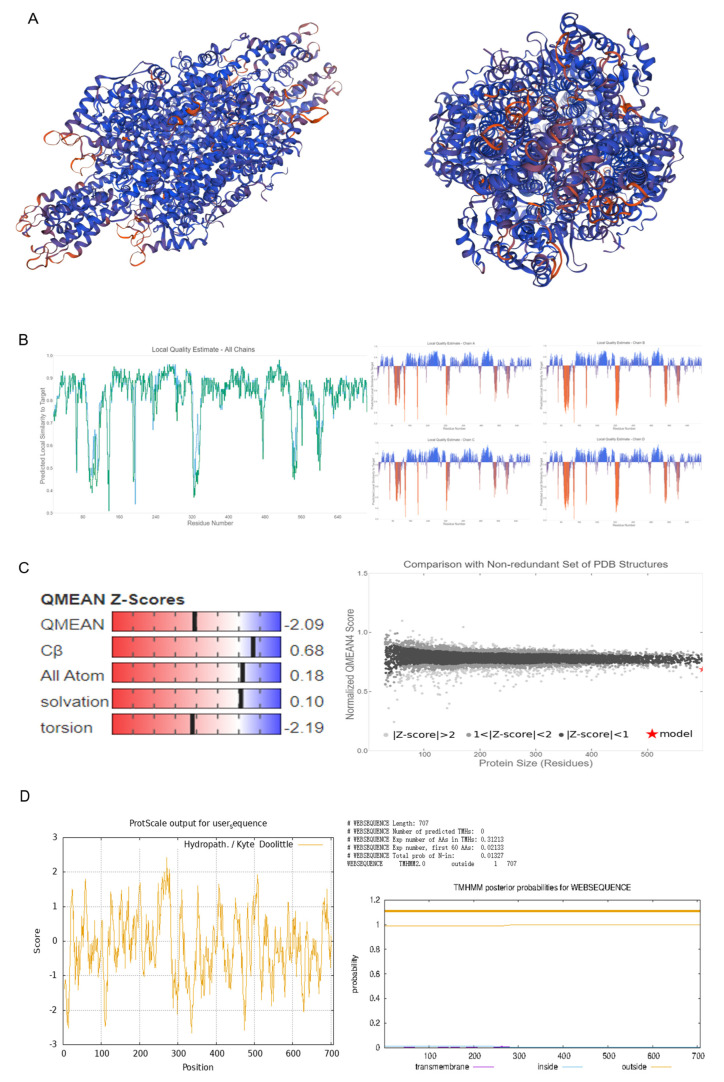
The prediction of tertiary structure and other key structural feature of CtPAL1 protein. (**A**) The tertiary structure model of CtPAL1 protein. (**B**) Quality assessment. (**C**) Matching analysis of CtPAL1 protein. (**D**) Hydrophobicity and transmembrane domain analysis.

**Figure 3 plants-14-03606-f003:**
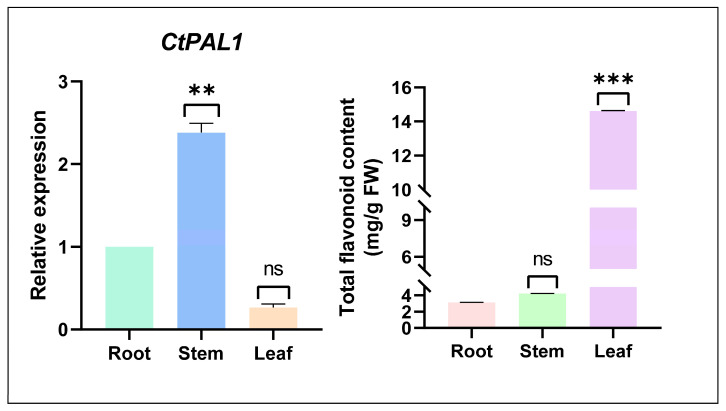
Expression pattern of *CtPAL1* and quantification of total flavonoid content in different tissues of safflower. Relative transcript levels of *CtPAL1* gene were quantified in roots, stems, and leaves of safflower, with root expression set as the baseline (value = 1). In parallel, total flavonoid contents were measured in the same tissues, showing higher accumulation in leaves compared to roots and stems. Asterisks denotes *** *p* < 0.05, and ** *p* < 0.01, Student’s *t*-test Data represent mean ± SD of three biological replicates.

**Figure 4 plants-14-03606-f004:**
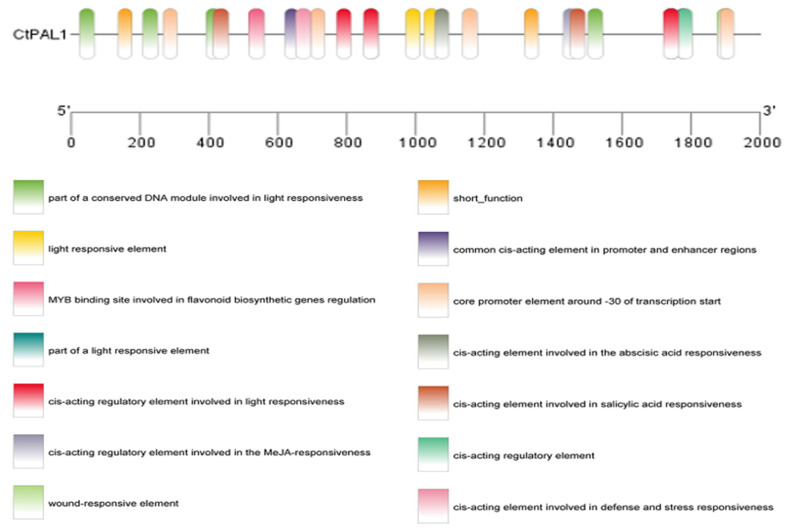
Cis-regulatory landscape of the *CtPAL1* promoter. Predicted cis-acting elements identified in the 2.0-kb upstream promoter region of CtPAL1. Functional motifs associated with hormone responsiveness (ABA, MeJA, SA, and GA), abiotic stresses (drought, salt, and UV-B), and light regulation are indicated. The presence of multiple stress- and defense-related motifs suggests that *CtPAL1* transcription is tightly integrated into environmental and hormonal signaling pathways.

**Figure 5 plants-14-03606-f005:**
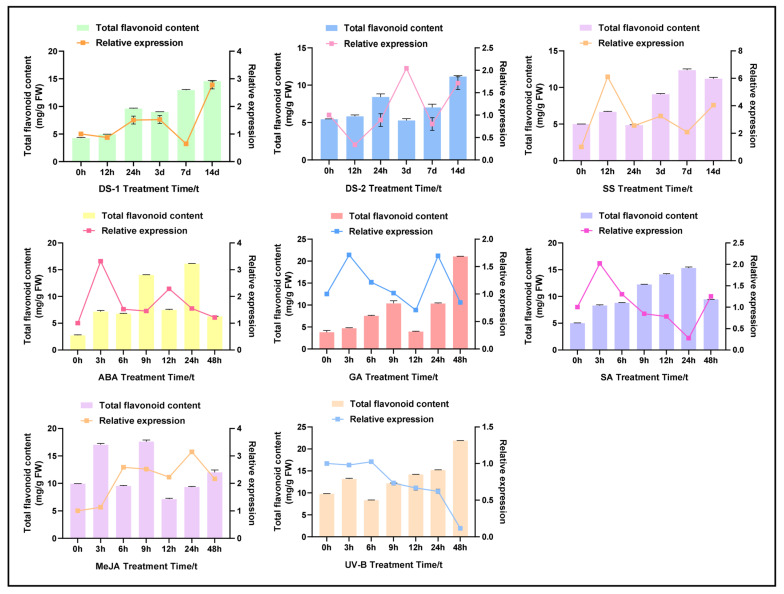
Induced expression dynamics of *CtPAL1* and total flavonoid accumulation under abiotic and hormonal treatments. The colored lines represent relative transcript abundance of *CtPAL1* in safflower subjected to drought (DS-1 and DS-2), salt (SS), ABA, MeJA, GA, SA, and UV-B treatments, using qRT-PCR at different time points, whereas the colored bars represent the corresponding changes in total flavonoid content under the same treatments and time points. Error bars represent the standard deviation of three biological replicates. Statistical significance with control samples are denoted as (*p* < 0.05, *p* < 0.01, Student’s *t*-test).

**Figure 6 plants-14-03606-f006:**
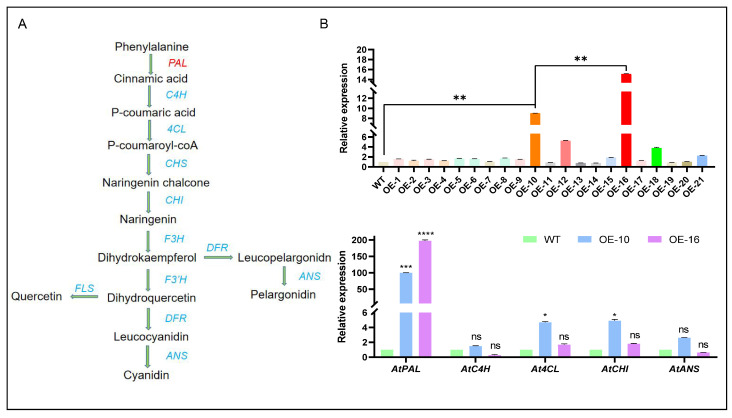
Transcript abundance of *CtPAL1* and other key flavonoid biosynthetic genes in *CtPAL1*-overexpression lines of Arabidopsis. (**A**) The representation of step-wise flavonoid biosynthetic pathway and their associated enzymes in plants. (**B**) Relative expression of *CtPAL1* in 21 transgenic lines (upper panel). The relative transcript abundance of four key genes of flavonoid biosynthetic pathway, namely, *At4CL*, *AtCHI*, *AtC4H*, and *AtANS*, using the two high-expression lines—OE10 and OE16 (lower panel). Statistical data are presented as mean ± SD for three independent biological replicates. Statistical significance relative to WT is shown with asterisks and distinct letters above the bars (**** *p* < 0.1, *** *p* < 0.05, ** *p*< 0.01 * *p* < 0.001, Student’s *t*-test).

**Figure 7 plants-14-03606-f007:**
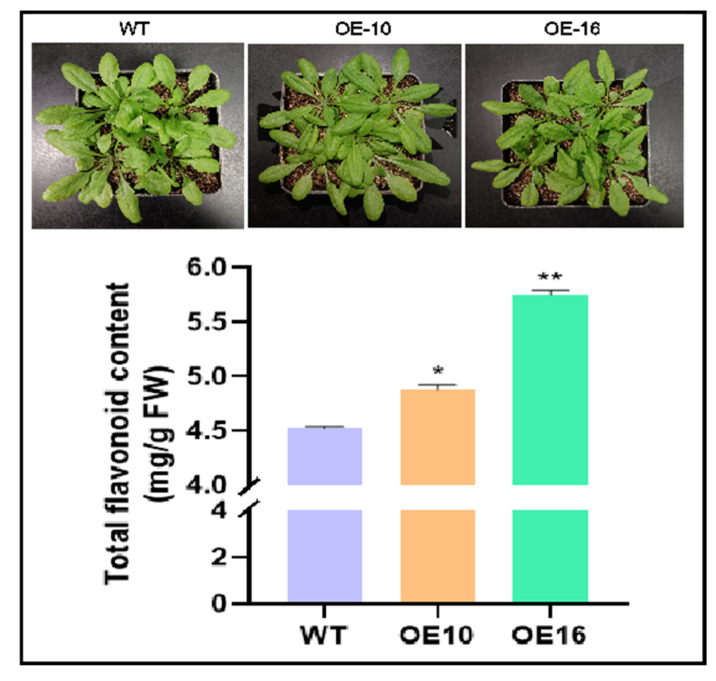
Enhanced flavonoid content and associated phenotypes in *CtPAL1*-overexpressing transgenic Arabidopsis. In the upper panel, we see the phenotypic representation of the WT and two transgenic lines (OE-10 and OE-16), demonstrating no obvious differences. In the lower panel, we see the quantitative data of total flavonoid content in WT and tow selected transgenic lines. Statistical data are presented as mean ± SD for three independent biological replicates. Statistical significance relative to WT is shown with asterisks and distinct letters above the bars (** *p* < 0.05, * *p* < 0.01, Student’s *t*-test).

**Figure 8 plants-14-03606-f008:**
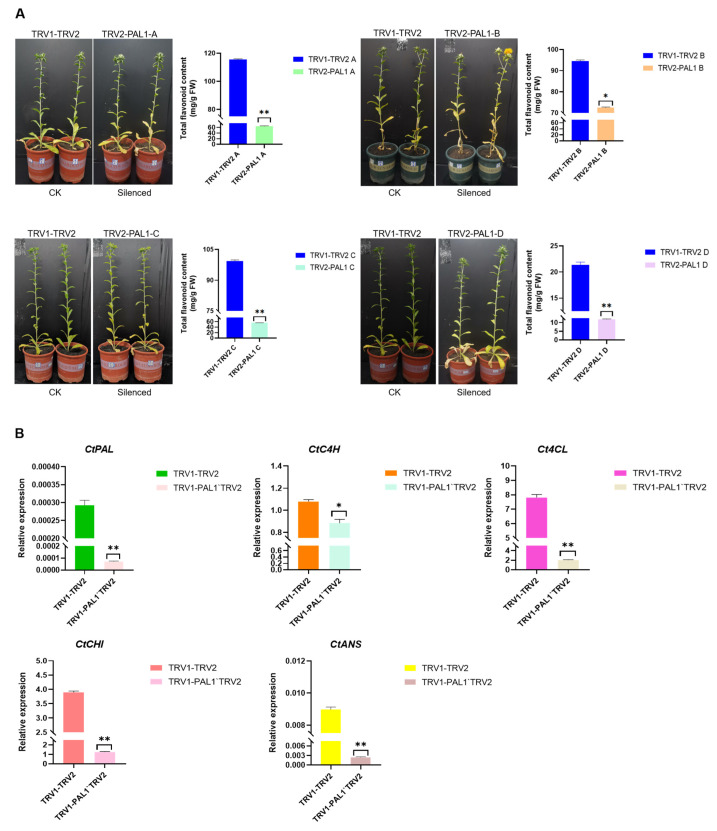
Phenotypic analysis, quantification of total flavonoid content, and expression analysis of *CtPAL1* and key flavonoid pathway genes in *CtPAL1* mutant. (**A**) Phenotypic variations and total flavonoid content in control and mutant plants two weeks after the infection. TRV 1 + TRV 2 served as the control, and TRV 1 + pTRV 2-*CtPAL1* as the mutant. (**B**) Relative expression levels of *CtPAL1* gene and four representative flavonoid pathway genes in *CtPAL1*-mutant plants. Data are shown as mean ± SD of three biological replicates. Asterisks indicate significant differences relative to the control (* *p* < 0.05, ** *p* < 0.01).

**Figure 9 plants-14-03606-f009:**
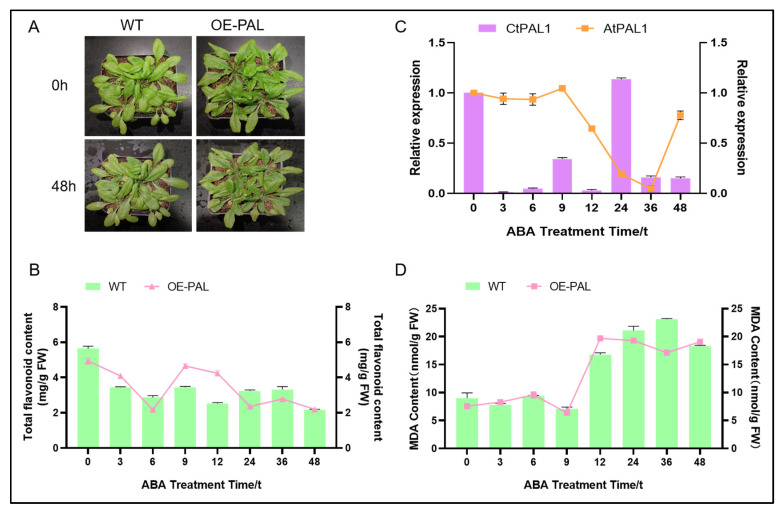
Phenotypic, expression, physiological, and biochemical responses of *CtPAL1*-overexpressing Arabidopsis under ABA treatment. (**A**) Phenotypic comparison of *CtPAL1*-overexpressing (OE-PAL) plants and WT under ABA elicitation. (**B**) Changes in total flavonoid content in WT and OE-PAL plants following ABA treatment. (**C**) Temporal expression pattern of *CtPAL1* and *AtPAL1* genes in transgenic lines under ABA treatment. (**D**) MDA content in WT and OE-PAL leaves during ABA treatment. Data represent means ± SD of three biological replicates.

**Figure 10 plants-14-03606-f010:**
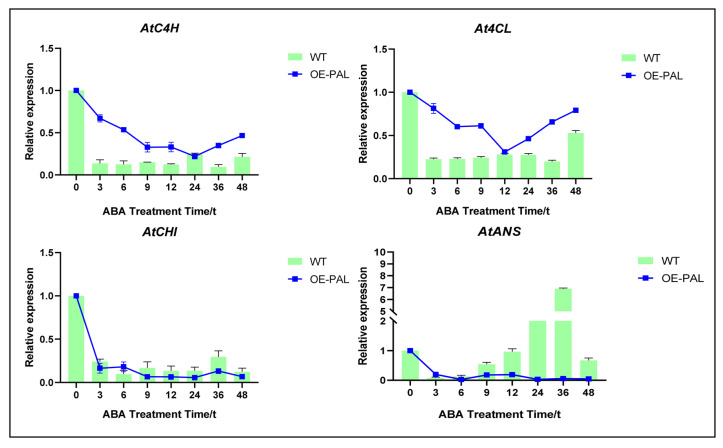
Comparative expression changes of the core flavonoid biosynthetic genes in *CtPAL1*-overexpressing transgenic and WT plants under the exposure of ABA treatment at different time points. The relative transcript abundance of the selected flavonoid genes including *AtC4H*, *At4CL*, *AtCHI*, and *AtANS* in *CtPAL1*-overexpressing transgenic and WT and plants were analyzed by qRT-PCR analysis. Statistical data represent means ± SD values of three independent biological replicates.

**Figure 11 plants-14-03606-f011:**
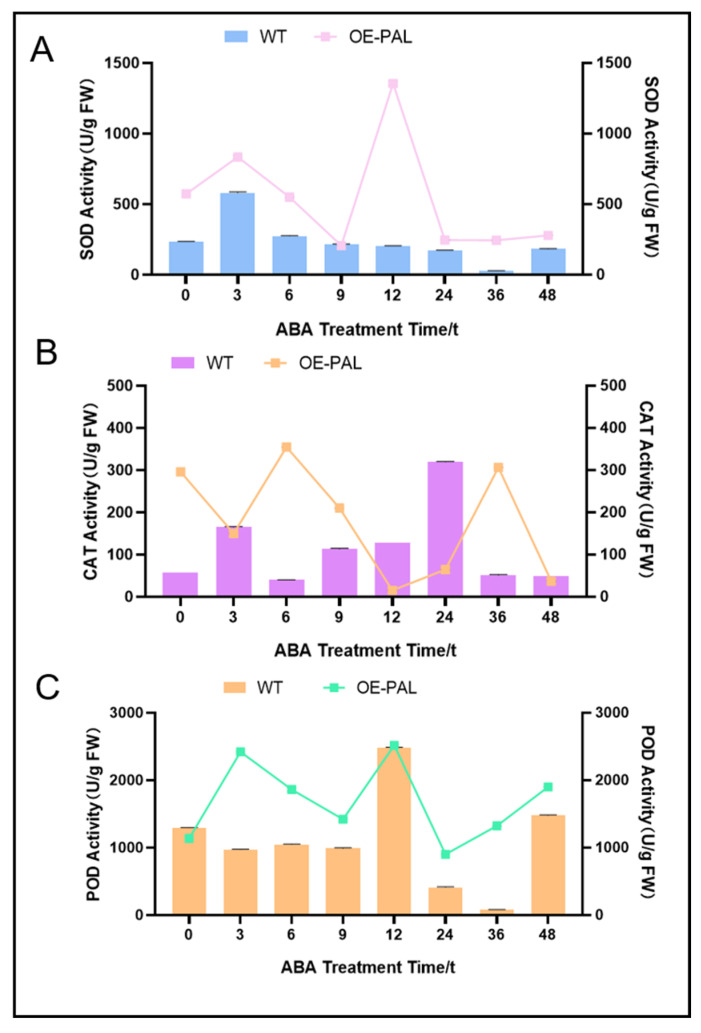
Improved antioxidant enzymatic activities in *CtPAL1*-overexpressing transgenic and WT plants under ABA condition at different time points. Enzymatic activities of (**A**) SOD, (**B**) CAT, and (**C**) POD. The data represent means ± SD values of three independent biological replicates.

**Figure 12 plants-14-03606-f012:**
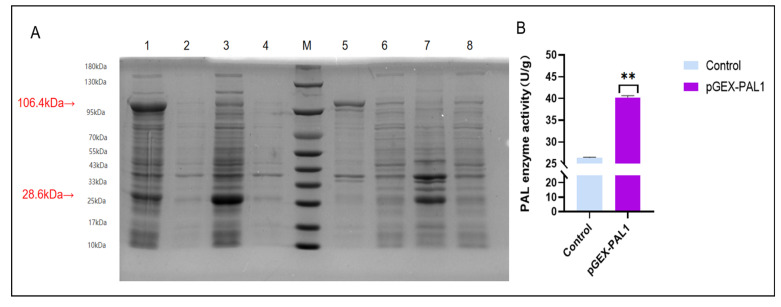
Protein expression, purification, and in vitro enzymatic activity of CtPAL1. (**A**) The analysis of SDS-PAGE using the recombinant expression vector harboring CtPAL1 in *Escherichia coli.* (**B**) The in vitro enzymatic activity showed that L-phenylalanine was catalyzed to trans-cinnamic acid using the purified CtPAL1 from bacterial cultures. Statistical data represents mean ± SD of three independent biological replicates. Asterisks indicate significant differences relative to the control (** *p* < 0.01).

## Data Availability

The original contributions presented in this study are included in the article/Appendix A. Further inquiries can be directed to the corresponding authors.

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
