# Peer review of "The Phenylpropanoid Gatekeeper CtPAL1 Coordinates ABA-Induced Flavonoid Biosynthesis and Oxidative Stress Tolerance in Safflower (*Carthamus tinctorius* L.)"

_plants, 2025, doi:10.3390/plants14233606_

Round 1

Reviewer 1 Report

Comments and Suggestions for Authors

The manuscript entitled “The Phenylpropanoid Gatekeeper CtPAL1 Coordinates ABA-Mediated Flavonoid Biosynthesis and Oxidative Stress Tolerance in Safflower (Carthamus tinctorius L)” is well written. However, the manuscript needs to address a few major questions before publishing in a reputed journal.

Why in figure 3, the relative expression of CtPAL is not aligning with flavonoid content? For example relative expression in the root is higher than leaf, while the flavonoid content is less than leaf. Expression is totally opposite to the flavonid content, while authors have reported that the CtPAL1 expression aligns with the flavonoid content.

The picture quality of figures 3 and 4 is very poor.

Figure 5. I hardly see the proper pattern of relative expression and flavonoid content. I think, flavonoid content is being regulated by some other dependent factors. CtPAL1 and flavonoids look like independent factors, which do not much affect each other.

Figure 7: No phenotypic differences in the OE 10 and OE 16 as compared to the WT of Arabidopsis, why?

Figure 8., I am surprised silencing is not affecting any phenotypic characteristics in plants. While the flavonoid content is decreasing by more than half compared to the control?

Author Response

Response to Reviewer 1

Comments and Suggestions for Authors

The manuscript entitled “The Phenylpropanoid Gatekeeper CtPAL1 Coordinates ABA-Mediated Flavonoid Biosynthesis and Oxidative Stress Tolerance in Safflower (Carthamus tinctorius L)” is well written. However, the manuscript needs to address a few major questions before publishing in a reputed journal.

Response:

We sincerely thank Reviewer 1 for the time, effort, and constructive feedback provided on our manuscript entitled. We truly appreciate the reviewer’s positive assessment that the manuscript is well written and acknowledge the thoughtful comments that have helped us to identify areas needing clarification and improvement. We have carefully addressed all the concerns, including the alignment of CtPAL1 expression with flavonoid content, the quality of figures, and the phenotypic responses of overexpression and silencing lines. The manuscript has been thoroughly revised to ensure better data presentation, improved discussion, and enhanced overall clarity. Our detailed, point-by-point responses are provided below.

Comment 1:

Why in Figure 3, the relative expression of CtPAL1 is not aligning with flavonoid content? For example, expression in the root is higher than leaf, while the flavonoid content is less than in the leaf. Expression is totally opposite to the flavonoid content, while authors have reported that the CtPAL1 expression aligns with the flavonoid content.

Response:

We appreciate the reviewer for this important observation. We apologize for not making it clear in our first version of the manuscript that the expression pattern presented in Figure 3 represents the basal tissue-specific expression of CtPAL1 under normal growth conditions, prior to any hormonal and any other stress treatment. We agree that the title of subsection 2.2 is contradictory and misleading. This apparent contradiction can be explained by the complex, multi-level regulation of the phenylpropanoid pathway, where transcript levels of a single upstream enzyme do not always predict downstream metabolite accumulation. CtPAL1 functions as a gatekeeper enzyme that channels phenylalanine into multiple branches of phenylpropanoid metabolism, including lignin, coumarins, stilbenes, and flavonoids. The relative flux into each branch is tissue-dependent, reflecting physiological priorities rather than PAL expression alone. Importantly, our subsequent experiments under ABA and abiotic stress treatments (Sections 2.3) revealed a coordinated induction of CtPAL1 expression and flavonoid accumulation, supporting that CtPAL1’s role in flavonoid biosynthesis is conditional and ABA-dependent, rather than constitutive under normal conditions. To clarify this interpretation, we have revised the manuscript text in section 2.2 (Lines 190–211).

Comment 2:

The picture quality of Figures 3 and 4 is very poor.

Response:

We have replaced Figures 3 and 4 with high-resolution pictures in the revised manuscript. (Please refer to updated Figures 3 and 4).

Comment 3:

Figure 5. I hardly see the proper pattern of relative expression and flavonoid content. I think flavonoid content is being regulated by some other dependent factors. CtPAL1 and flavonoids look like independent factors, which do not much affect each other.

Response:

In response to the reviewer's observation, we agree that the regulatory landscape is multifaceted. Our findings, as concluded in the figure 5, indicate that certain treatments like salt and SA trigger an early CtPAL1 response that decouples from later flavonoid accumulation, suggesting post-transcriptional control. In contrast, GA and UV-B promote flavonoid biosynthesis through largely CtPAL1-independent pathways. From this spectrum of responses, ABA emerged as the most significant factor because it alone exhibited a clear, coordinated relationship where transient peaks in CtPAL1 expression were logically followed by significant flavonoid accumulation. This compelling temporal increase provided us a strong rationale for selecting ABA to delineate the precise mechanistic connection in subsequent experiments.

Comment 4:

Figure 7: No phenotypic differences in the OE10 and OE16 as compared to the WT of Arabidopsis, why?

Response:

We thank the reviewer for noticing this. The overexpression of CtPAL1 in Arabidopsis did not cause obvious Phenotypic changes under normal growth conditions, which is consistent with reports for other PAL genes. PAL activity is more functionally evident under stress or hormone induction rather than in basal developmental stages. In our subsequent experiments, OE lines under ABA elicitations also exhibited improved antioxidant capacity, and enhanced expression of key flavonoid  biosynthetic genes compared to WT, confirming the functional differences but not phenotypic. This clarification has been added in the revised manuscript (Highlighted in yellow, Lines 359–376.)

Comment 5:

Figure 8: I am surprised silencing is not affecting any phenotypic characteristics in plants, while the flavonoid content is decreasing by more than half compared to the control?

Response:

We sincerely apologize for the misunderstanding caused by the lack of proper labeling in Figure 8, which may have led to confusion in interpreting the results. There are indeed slight phenotypic changes occurred after injecting TRV vector into safflower, however, as mentioned earlier, CtPAL1 is mainly associated with functional roles, which is typically more evident under stress or hormone-induced conditions rather than during basal developmental stages. We have revised the figure labelling for clarity (See updated fig. 8).

Reviewer 2 Report

Comments and Suggestions for Authors

This manuscript presents a comprehensive functional characterization of CtPAL1, a phenylalanine ammonia-lyase gene in safflower. By combining phylogenetic analysis, structural predictions, transgenic Arabidopsis overexpression, virus-induced silencing in safflower, enzymatic assays, and physiological/biochemical assessments, the authors convincingly demonstrate that CtPAL1 acts as a regulatory hub connecting ABA signaling to flavonoid biosynthesis and stress tolerance.

The study is timely and relevant, as flavonoids play central roles in both plant stress resilience and human health benefits, while PAL enzymes serve as critical nodes in phenylpropanoid metabolism. The experimental design is ambitious, integrating molecular biology, genetics, and biochemistry. The results are clearly presented, well-supported by figures and supplementary data, and the discussion appropriately contextualizes the findings within the framework of previous studies. Overall, this study makes a valuable contribution to the field of plant physiology and secondary metabolism.

The promoter analysis and hormone treatments convincingly establish the responsiveness of CtPAL1 to ABA, and the functional assays show enhanced antioxidant enzyme activity and flavonoid accumulation. The observed improvements in oxidative stress tolerance and root growth under salt and ABA stress in transgenic plants highlight practical implications for crop resilience.

Major Comments
1. ROS and polyphenol metabolism
The major focus of this study should be the ROS production triggered by ABA signaling under stress conditions, which plays a central role in oxidative stress responses. The results clearly show that overexpression of CtPAL1 reduces MDA content while increasing SOD, CAT, and POD activities. However, the discussion does not adequately address the involvement of flavonoids and other polyphenolic secondary metabolites. These compounds are important substrates for guaiacol peroxidases (such as horseradish peroxidase, HRP) in detoxifying H2O2, similar to the function of ascorbate peroxidase (APX) with ascorbate as an electron donor. It appears that the study measured a mixture of different types of peroxidase activities. I strongly suggest that the authors expand the discussion to consider both the non-enzymatic antioxidant functions of flavonoids and their enzymatic role in H2O2 detoxification.

See for example:
DOI: 10.1016/S0300-483X(02)00196-8
DOI: 10.1016/S1360-1385(97)82730-6
DOI: 10.5772/intechopen.95627

2. Novelty
While PAL genes have been extensively studied in other species, the novelty here lies in the ABA-mediated regulation in safflower. The discussion should emphasize more clearly how CtPAL1 differs functionally from other PAL isoforms in safflower and why it is designated the “gatekeeper.”

3. CtPAL1 regulatory crosstalk with other pathways
The interaction between CtPAL1 and downstream pathway genes (C4H, 4CL, CHI, ANS) is described primarily at the expression level. To strengthen the claim of regulatory crosstalk, the authors should consider providing additional mechanistic insights (e.g., transcription factor binding studies, metabolite flux analyses).

4.    Data interpretation
In some cases—such as the response to UV-B treatments or the biphasic expression of CtPAL1 under ABA stress—the interpretation appears speculative. These sections should be revised to acknowledge alternative explanations and limitations more explicitly.

Minor Comments
1.  The introduction could be streamlined by reducing the pharmacological details of safflower and focusing instead on the gaps in PAL functional research.

2. The supplementary tables are useful but should be explicitly cited in the Results section to guide readers more effectively.

Author Response

Reviewer 2

Comments and Suggestions for Authors

This manuscript presents a comprehensive functional characterization of CtPAL1, a phenylalanine ammonia-lyase gene in safflower. By combining phylogenetic analysis, structural predictions, transgenic Arabidopsis overexpression, virus-induced silencing in safflower, enzymatic assays, and physiological/biochemical assessments, the authors convincingly demonstrate that CtPAL1 acts as a regulatory hub connecting ABA signaling to flavonoid biosynthesis and stress tolerance.

The study is timely and relevant, as flavonoids play central roles in both plant stress resilience and human health benefits, while PAL enzymes serve as critical nodes in phenylpropanoid metabolism. The experimental design is ambitious, integrating molecular biology, genetics, and biochemistry. The results are clearly presented, well-supported by figures and supplementary data, and the discussion appropriately contextualizes the findings within the framework of previous studies. Overall, this study makes a valuable contribution to the field of plant physiology and secondary metabolism.

The promoter analysis and hormone treatments convincingly establish the responsiveness of CtPAL1 to ABA, and the functional assays show enhanced antioxidant enzyme activity and flavonoid accumulation. The observed improvements in oxidative stress tolerance and root growth under salt and ABA stress in transgenic plants highlight practical implications for crop resilience.

Response:

We sincerely thank Reviewer 2 for the detailed and insightful evaluation of our manuscript and for recognizing the strength, relevance, and novelty of our study. We are grateful for the reviewer’s encouraging remarks highlighting the comprehensive experimental design, clear presentation, and significance of our findings in understanding the ABA-mediated regulation of CtPAL1 in safflower. We have thoroughly revised the manuscript to address all major and minor comments, expanded the discussion where appropriate, and refined the interpretation of results to improve scientific depth and coherence. Detailed, point-by-point responses to each comment are provided below.

Major Comments

  1. ROS and polyphenol metabolism

The major focus of this study should be the ROS production triggered by ABA signaling under stress conditions, which plays a central role in oxidative stress responses. The results clearly show that overexpression of CtPAL1 reduces MDA content while increasing SOD, CAT, and POD activities. However, the discussion does not adequately address the involvement of flavonoids and other polyphenolic secondary metabolites. These compounds are important substrates for guaiacol peroxidases (such as horseradish peroxidase, HRP) in detoxifying H2O2, similar to the function of ascorbate peroxidase (APX) with ascorbate as an electron donor. It appears that the study measured a mixture of different types of peroxidase activities. I strongly suggest that the authors expand the discussion to consider both the non-enzymatic antioxidant functions of flavonoids and their enzymatic role in H2O2 detoxification.

See for example:

DOI: 10.1016/S0300-483X(02)00196-8

DOI: 10.1016/S1360-1385(97)82730-6

DOI: 10.5772/intechopen.95627

Response:

We appreciate this valuable suggestion. Based on your suggestion, we have revised the discussion to highlight the dual antioxidant role of flavonoids, referencing and citing the suggested literature. The revised section now explains that flavonoids act as both non-enzymatic ROS scavengers and enzymatic substrates for guaiacol peroxidases involved in H₂O₂ detoxification. Please see the revision (Highlighted in yellow, Lines 601–620.)

  1. Novelty

While PAL genes have been extensively studied in other species, the novelty here lies in the ABA-mediated regulation in safflower. The discussion should emphasize more clearly how CtPAL1 differs functionally from other PAL isoforms in safflower and why it is designated the “gatekeeper.”

Response:

We thank the reviewer for this excellent point. We have revised the discussion to explicitly differentiate CtPAL1 from other PAL isoforms based on its ABA responsiveness, co-expression with flavonoid biosynthetic genes, and stress-induced regulation. (Highlighted in yellow, Lines 621–640 and 641-671).

  1. CtPAL1 regulatory crosstalk with other pathways

The interaction between CtPAL1 and downstream pathway genes (C4H, 4CL, CHI, ANS) is described primarily at the expression level. To strengthen the claim of regulatory crosstalk, the authors should consider providing additional mechanistic insights (e.g., transcription factor binding studies, metabolite flux analyses).

Response:

We have expanded this section to include a discussion of possible transcriptional and metabolic interactions, referencing available regulatory studies on PAL-linked transcription factors  (Highlighted in yellow, Lines 671–710).

  1. Data interpretation

In some cases—such as the response to UV-B treatments or the biphasic expression of CtPAL1 under ABA stress—the interpretation appears speculative. These sections should be revised to acknowledge alternative explanations and limitations more explicitly.

 Response:

We thank the reviewer for this constructive note. We have revised these sections to tone down speculative language and added statements acknowledging possible alternative regulatory mechanisms, including transient ABA feedback loops and ROS signaling.

Minor Comments

  1. The introduction could be streamlined by reducing the pharmacological details of safflower and focusing instead on the gaps in PAL functional research.

Response:

The introduction has been revised by focusing more on the knowledge gap in PAL functional research rather than the pharmacological aspects of safflower. (Highlighted in yellow, Lines 43–60).

  1. The supplementary tables are useful but should be explicitly cited in the Results section to guide readers more effectively.

Response:

We have added explicit references to all supplementary tables within the Results section for clarity.

(Highlighted in yellow, Lines 102-106, and 116–119).

Round 2

Reviewer 1 Report

Comments and Suggestions for Authors

Authors have incorporated all the suggestions properly. I recommend publishing this manuscript in its current version.